# A Novel Synthetic Strategy for Preparing Polyamide 6 (PA6)-Based Polymer with Transesterification

**DOI:** 10.3390/polym11060978

**Published:** 2019-06-03

**Authors:** Shengming Zhang, Jingchun Zhang, Lian Tang, Jiapeng Huang, Yunhua Fang, Peng Ji, Chaosheng Wang, Huaping Wang

**Affiliations:** 1State Key Laboratory for Modification of Chemical Fibers and Polymer Materials, Key Laboratory of Textile Science & Technology (Ministry of Education), College of Materials Science and Engineering, Donghua University, Shanghai 201620, China; SMZhang_dhu@163.com (S.Z.); zjc2633511565@163.com (J.Z.); marstanglian@sina.com (L.T.); jphuang0818@163.com (J.H.); fyh96730@163.com (Y.F.); wanghp@dhu.edu.cn (H.W.); 2Co-innovation center for textile industry, Shanghai 201620, China

**Keywords:** synthetic methodology, polyamide 6, melt polymerization, polyesteramide, transesterification, polyamide 6 (PA6)-based polymer

## Abstract

In the polymerization of caprolactam, the stoichiometry of carboxyl groups and amine groups in the process of melt polycondensation needs to be balanced, which greatly limits the copolymerization modification of polyamide 6. In this paper, by combining the characteristics of the polyester polymerization process, a simple and flexible synthetic route is proposed. A polyamide 6-based polymer can be prepared by combining caprolactam hydrolysis polymerization with transesterification. First, a carboxyl-terminated polyamide 6-based prepolymer is obtained by a caprolactam hydrolysis polymerization process using a dibasic acid as a blocking agent. Subsequently, ethylene glycol is added for esterification to form a glycol-terminated polyamide 6-based prepolymer. Finally, a transesterification reaction is carried out to prepare a polyamide 6-based polymer. In this paper, a series of polyamide 6-based polymers with different molecular weight blocks were prepared by adjusting the amount and type of dibasic acid added, and the effects of different control methods on the structural properties of the final product are analyzed. The results showed that compared with the traditional polymerization method of polyamide 6, the novel synthetic strategy developed in this paper can flexibly design prepolymers with different molecular weights and end groups to meet different application requirements. In addition, the polyamide 6-based polymer maintains excellent mechanical and hygroscopic properties. Furthermore, the molecular weight increase in the polyamide 6 polymer is no longer dependent on the metering balance of the end groups, providing a new synthetic route for the copolymerization of polyamide 6 copolymer.

## 1. Introduction

Due to its high strength, good fatigue resistance, good water absorption, and chemical stability, polyamide 6 is one of the most widely-used engineering thermoplastics. It is also often referred to as nylon 6 or polycaprolactam in industrial production and applications. Polyamide 6 is mainly obtained by hydrolytic polymerization of caprolactam and anionic polymerization. Among these methods, caprolactam hydrolysis polymerization is the main approach for the mass production of polyamide 6 due to its polymerization stability and controllability. During the hydrolysis and polymerization of caprolactam, the molecular chain is grown by dehydration condensation of terminal carboxyl groups and amine groups. Therefore, it is necessary to maintain the stoichiometric balance of carboxyl groups and amine groups during the polycondensation process. In the copolymerization modification of polyamide 6, the modified polymer component added tends to affect the end group balance in the system, leading to a decrease in the degree of polymerization. Therefore, to achieve copolymerization modification of polyamide 6, the existing research mainly involves copolymerization modification by adding different chain extenders [1,2,3,4,5].

The addition of a chain extender for copolymerization modification mainly relies on an esterification reaction of a terminal carboxyl group on a polyamide 6 molecule with a glycol or a polyether ester or an amidation reaction with a diamine. Camille Bakkali-Hassani et al. [6] prepared a polyamide 6-based copolymer with a benzene structural unit by anionic polymerization using ethyl 4-aminobenzoate as a chain extender. Shumin Peng et al. [7] prepared a polyamide 6-based copolymer by anionic polymerization using ethylenediamine as a chain extender. Thibault Cousin et al. [8] prepared a bio-based polyamide copolymer by adding 2,5-furandicarboxylic acid during the hydrolysis reaction of caprolactam. However, regardless of which polymerization method is used, the addition of a dibasic acid or a diamine as a chain extender changes the reactivity between different substances in the reaction system, thereby destroying the stoichiometric balance between the terminal carboxyl group and the terminal amino group. Finally, the molecular chain is blocked, limiting the growth of molecular weight. Compared with the small-molecule chain extenders modified for polyamide 6, the use of polyether ester as a modifier has a milder polymerization effect and is also more suitable for the caprolactam hydrolysis polymerization system. Li-Kun Xiong et al. [9] prepared cationic dyeable polyamide 6 fibre by using 5-sulfoisophthalic acid monosodium salt as a blocking agent and melt polymerization with polyethylene glycol. Juncheng Huang and Ruchao Yuan et al. [10,11] prepared a polyamide 6 elastomer by using adipic acid and terephthalic acid as blocking agents and esterification of polytetrahydrofuran under melt conditions, respectively. However, in the process of caprolactam hydrolysis polymerization, approximately 8%–10% of oligomers are inevitably formed, also known as hot water extractables [12,13]. Usually, the polyamide 6 slices are extracted with boiling water or superheated water before the product leaves the factory; otherwise, the quality of the material is affected. In the preparation of polyamide 6 matrix materials in the laboratory, we usually use the same treatment methods to ensure the processing performance of the materials [7,14]. After modification copolymerization with a chain extender, an ester group and an ether group are usually introduced in a macromolecular segment, and the two chemical groups are easily broken during high-temperature hydrothermal treatment to cause a decrease in molecular weight [15,16]. This phenomenon has become another problem encountered when using chain extenders for polyamide 6 modification.

Compared with the synthetic route for polyamide 6, the synthetic route and modification method for polyethylene terephthalate (PET) are more flexible. At present, terephthalic acid (PTA) and ethylene glycol (EG) are often used in industry to form 2-(hydroxyethyl terephthalate) (BHET) by an esterification reaction, and PET is prepared by transesterification of BHET under high-temperature and vacuum conditions. There is a difference between PET synthesis and polyamide 6 synthesis. The chain growth mode is based on transesterification. The ends of the ester compound are the same and can react with each other. At the same time, transesterification can also occur between the oligomer and the polymer, enhancing the stability of the reaction [17,18,19,20]. The modification of PET mainly involves adding a kind of glycol, an oligomer of a glycol, a dibasic acid, an aromatic dibasic acid, or a polyether ester to the system by an esterification reaction or a transesterification reaction. Compared with the copolymerization modification of polyamide 6, the copolymerization modification of PET has the characteristics of high modifier addition content and diversified addition types, and it is easier to design different molecular segments with different complex functions [21,22,23,24]. It is worth mentioning that copolymer-modified cationic dyeable PET (CDP/ECDP) has achieved industrial production, indicating that the synthetic route for PET has good stability in copolymerization modification [25].

Inspired by the synthesis route of PET, we propose a simple and flexible melt polymerization route for the preparation of PA6-based polymers. First, with different molar ratios of adipic acid, PTA and other dibasic acid blocking agents are added to the caprolactam hydrolysis polymerization to prepare a polyamide 6-based prepolymer terminated by a carboxyl group at both ends. Subsequently, ethylene glycol is added for esterification to prepare a polyamide 6-based ethylene glycol ester. Finally, the transesterification reaction is carried out to obtain a long-chain polymer. In the molecular design process, different functionalization effects can be achieved by changing the blocking agent, which provides a new synthetic route for the copolymerization modification of the polyamide 6-based polymer. In addition, the polyamide 6-based polymer synthesized by this paper reduces the hot water extractable content by 75%. The excellent mechanical properties and moisture absorption properties of the polymer can be maintained by direct processing without hydrothermal extraction, which is also difficult to achieve by other synthetic routes for copolymerization modification of polyamide 6.

## 2. Experimental 

### 2.1. Materials

PA6 chips were purchased from CHTC Sinofiber Wuxi Co. Ltd. (Wuxi, China). Caprolactam was purchased from BASF SE (Shanghai, China). Titanium glycolate (Ti(OCH_2_CH_2_O)_2_) was purchased from Xinfengming Group (Zhejiang, China). Ethylene glycol (EG), terephthalic acid (PTA), adipic acid, deuterated concentrated sulfuric acid (D_2_SO_4_), and 98% concentrated sulfuric acid (H_2_SO_4_) were purchased from Sinopharm Chemical Reagent Co. Ltd. (Shanghai branch, China). All the materials were analytically pure.

### 2.2. Sample Preparation

#### 2.2.1. Synthesis of Polyamide 6 (PA6)-based Polymer

The synthetic route for the PA6-based polymer is shown in Scheme 1. All reactions were carried out in a 5 L stainless steel reactor with a vacuum tube, vacuum pump and nitrogen bottle. Specific steps are as follows.

• Step 1: Synthesis of PA6 with binary carboxyl end groups

In a typical polymerization procedure, a mixture of CPL (1695.0 g, 15 mol), adipic acid (146.14 g, 1 mol), and deionized water (44.0 g, 2.44 mol) was transferred into a vessel and stirred at 4 bar gas pressure and 250 °C for 3 h. The pressure was then slowly decreased to atmospheric pressure, and then nitrogen was used to remove excess water. The reaction was kept at atmospheric pressure for another 1 h, and the products were kept in the vessel for the next experiment.

• Step 2: Synthesis of PA6 with binary glycol ester end groups (prePA6)

In a typical polymerization procedure, EG (59.53 g, 0.96 mol) and Ti(OCH_2_CH_2_O)_2_ (0.136 g, effective content of 1%) were introduced into the reactor. The reaction was maintained at 2 bar gas pressure and 240 °C for 2.5 h. The pressure was then gradually reduced to atmospheric pressure for 1 h. The products were kept in the vessel for the next experiment. The prepolymer molecular weights were controlled at 2000, 3000, 4000, 5000, and 6000, and the samples were coded as prePA6-2K, prePA6-3k, prePA6-4k prePA6-5k, and prePA6-6k, respectively. The prepolymer with its molecular weight controlled at 4000 and terminated by PTA was coded as prePA6-p-4k.

• Step 3: Synthesis of the polyamide 6 (PA6)-based polymer (polyesteramide, PEA)

In a typical polymerization procedure, the pressure was then gradually reduced to less than 100 Pa, and the temperature was increased to 280 °C to remove the generated EG. The reaction was kept at this pressure until the specified power was reached. Then, the resulting products were condensed in cool water. Polymers obtained by polycondensation of prepolymers with different molecular weights were coded as PEA-2k, PEA-3k, PEA-4k, PEA-5k, PEA-6k, and PEA-p-4k.

#### 2.2.2. Preparation of PA6/PEA Plastic Splines

For preparation of dumbbell-shaped samples by hot pressing, the pelletized samples were injected using an injection molding machine to obtain the test specimens for each characterization technique. The temperature of the cylinders was 250 °C, and the mold temperature was 60 °C. The PEA chips are processed without hydrothermal extraction. PA6 chips are processed by hydrothermal extraction before the chips leave the factory.

### 2.3. Measurements

#### 2.3.1. ^1^H-NMR and FTIR

^1^H-NMR experiments were performed on an AVANCE-600 (Switzerland) spectrometer. D_2_SO_4_ was used as the solvent for prePA6 and PEA with a test temperature of 25 °C and the frequency of 600 MHz. FTIR spectroscopy is performed with a Nicolet 6700 Fourier transform infrared spectrometer (MA, USA). The scanning range is 4000~600 cm^−1^.

#### 2.3.2. Relative Viscosity

The prePA6 samples were tested at (25 ± 0.1) °C according to GB/T 12006.1-2006 using a Ubbelohde viscometer with a capillary diameter of 1.03 mm. The solvent was 98% concentrated sulfuric acid. The relative molecular mass (*M_η_*) was calculated according to the following formula (1) and (2).
(1)$$\eta_{r}=\frac{t}{t_{0}}$$
(2)$$M_{\eta }=\left(\eta_{r}-1\right)\times 11500$$
In the formula:*t_0_*—the outflow time of the pure solvent, s;*t*—the outflow time of the concentrated sulfuric acid solution of PA6, s.

#### 2.3.3. Oligomer Content

The oligomer content of the prePA6 and PEA samples were tested by hydrothermal extraction. A sample of 20 g ± 0.5 g was dried in a vacuum oven for 8 h. The dried chips were weighed and placed in a constant temperature water bath of ≥97 °C for 8 h. Then, the water was removed by a centrifuge, the sample was dried in a vacuum oven at 120 °C for 8 h, the chips were weighed, and the mass was reduced to the weight of the oligomers. The experiment was repeated 5 times to obtain an average value.

#### 2.3.4. Detection of Amino and Carboxylic End Groups

The concentration of the amino groups [–NH_2_] and the carboxyl groups [–COOH] of the unesterified prePA6 was tested by titration. A sample of 0.25 ± 0.01 g was dissolved in 15 mL of a 75:25 volume mixture of trifluoroethanol and ethanol. The concentration of the amino end group was then tested using a standard hydrochloric acid solution (0.002 mol L^−1^), and the concentration of the carboxyl end group was tested using a standard sodium hydroxide solution (0.02 mol L^−1^). The experiment was repeated 5 times to obtain an average value.

#### 2.3.5. Gel Permeation Chromatography, GPC

The PEA molecular weight (*Mn* and *Mw*) and polymer dispersity index (PDI) were tested by a GPC equipped with a refractive index detector and a PL gel column (5 μm mixed-C). With 1,1,1,3,3,3-hexafluoro-2-propanol as the eluent, the flow rate was 1 mL min^−1^. A calibration curve is generated using polystyrene as a standard.

#### 2.3.6. Differential Scanning Calorimetry (DSC), and Thermogravimetric Analysis, TGA

DSC measurements were performed on a TA-Q20 differential scanning calorimeter apparatus calibrated with an indium standard. Samples of approximately 5 mg were heated and cooled at a rate of 20 °C min^−1^ under a nitrogen atmosphere. The first cooling scan and second heating scan were selected for studying the thermal properties of the samples. TGA was performed on a German 209-F1 thermogravimetric analyzer under a nitrogen atmosphere at a heating rate of 20 °C/min from 30 to 600 °C.

#### 2.3.7. Wide-angle X-ray Diffraction, WAXD

WAXD data were collected on a D/max-2550 VB X-ray diffractometer (Rigaku, Japan) equipped with a filtered Cu Kα radiation source (wavelength λ = 1.542 Å), and the diffraction patterns were recorded in a range of 2θ = 5° to 60°.

#### 2.3.8. Scanning Electron Microscope, SEM

SEM images were collected on by S-4800 field-emission scanning electron microscope produced by Hitachi G. The samples were treated with gold spray.

#### 2.3.9. Tensile Properties

Tensile property data were acquired at room temperature using a Changchun Kexin WDW3020 electronic universal testing machine. The PEA chips were processed into a dumbbell-shaped spline by injection molding. The tensile speed was 10 mm min^−1^, and the spline spacing was 50 ± 0.5 mm.

#### 2.3.10. Chips Saturated Water Absorption

The PA6 and PEA samples (20 ± 0.5 g) were dried in a drying oven for 12 h to a constant weight. The chips were removed, immersed in a beaker containing deionized water, and soaked for 48 h. The surface moisture was dried, and the sample was weighed. The weight increase is the chip saturated water absorption rate.

## 3. Results and Discussion

### 3.1. Preparation and Characterization of prePA6

As shown in Scheme 1, in the first step of caprolactam hydrolysis polymerization, PA6 segments of different number average molecular weights were prepared by adding a dicarboxylic acid in different molar ratios relative to caprolactam. The dicarboxylic acid can be one or several aliphatic compounds or aromatic compounds. In this paper, adipic acid and PTA were used as representatives of these two kinds of compounds. The molar ratio of caprolactam to dicarboxylic acid [PCL]/[dicarboxylic acid] with different molecular chain lengths of PA6 and the content of terminal amino groups and terminal carboxyl groups are shown in Table 1.

In the second step, ethylene glycol was added as a blocking agent to the prepolymer sample in an amount 2.2 times the carboxyl group content in the PA6 segment. The esterification rate of the polyamide 6 prepolymer is represented by the mass of water produced. The esterification yield, crystallization temperature (*Tc*), melting temperature *(Tm*), and relative molecular weight (*Mη*) of the prepared prePA6 samples are shown in Table 1. Differential scanning calorimetry (DSC) results of the prePA6 samples are shown in Appendix A.

The chemical structure of prePA6 was characterized by ^1^H-NMR. Figure 1a shows the ^1^H-NMR spectrum of prePA6-2k with D_2_SO_4_ as the solvent. The chemical shifts at 3.88 ppm, 3.06 ppm, 2.12 ppm, 2.06 ppm, and 1.80 ppm were from the δ_5_, δ_1_, δ_4_, δ_2_, and δ_3_ hydrogen atoms of the polyamide 6 segment, respectively. The chemical shift at 3.12 ppm is the hydrogen on the carbon that is linked to the ester bond from δ_1’_. The chemical shifts at 2.93 ppm and 2.19 ppm are considered the characteristic hydrogen shifts of adipic acid at δ_6_, δ_7_. The chemical shifts at 4.37 ppm, 4.45 ppm, 4.69 ppm, and 4.79 ppm are considered the hydrogen atoms on the ethylene glycol segment. Since the carboxyl groups at both ends of the PA6 segment can be reacted with ethylene glycol, there are three different linkage modes: A-A, A-B, and B-B. Similar to the esterification process between PTA and EG, after the esterification reaction is completed, prePA6 with a polymerization degree of 1 to 4 is formed, and the characteristic peaks of hydrogen atoms correspond to δ_8_, δ_9_, δ_10_, and δ_11_, respectively. The molecular weight of the PA6 segment can be calculated from the ratio of the integral area of the hydrogen shift characteristic of ε-aminocaproic acid to that of the hydrogen shift of adipic acid [8]. The calculation results are shown in Table 1.
(3)$$M_{nprePA6}=\left(I_{5}/I_{6}\times 2\right)\times 113+146$$

In the formula (3), *I_5_* and *I_6_* are the characteristic peak integral areas corresponding to δ_5_ and δ_6_ in the figure, respectively. Number 133, 146 represent the relative molecular mass of caprolactam and adipic acid, respectively.

### 3.2. Preparation and Characterization of PEA

In this paper, the transesterification reaction of different molecular weight prePA6 samples is used as the chain growth principle of melt polymerization. It is worth noting that under high vacuum (below 100 Pa), not only the ethylene glycol produced by the transesterification reaction but also the oligomers (mainly cyclic dimers) produced during the caprolactam hydrolysis polymerization are removed. A certain preheating device should be to prevent condensation of the oligomer on the vacuum pipe. Figure 1b shows the ^1^H-NMR spectrum of PEA-2k prepared by polycondensation with aliphatic adipic acid as a blocking agent. The chemical shifts and the ratio of the characteristic peak integral area are basically the same as those of prePA6, indicating that there is no structural change in the PA6 segment before and after the transesterification, which makes the effect of amide-transesterification negligible. The hydrogen shift ratio of the ethylene glycol segment was significantly reduced, which shows that chain growth is achieved by transesterification.

Figure 1c shows the ^1^H-NMR spectrum of PEA-p-4k prepared by polycondensation of prePA6-p-4k with aromatic terephthalic acid as a blocking agent. The chemical shift at 8.2~8.6 ppm is the characteristic peak of the hydrogen atoms in the benzene ring [11]. Because of the high electron cloud density of benzene, the chemical environment of the hydrogen on the adjacent ε-aminocaproic acid is greatly influenced, leading to a high-field shift in this hydrogen. As the same as Figure 1a, the δ_1’_ is the hydrogen on the carbon that is linked to the ester bond. The integrated areas of the characteristic hydrogen peaks at the same position were combined when calculating the number average molecular weight. In addition, the hydrogen in PTA also exhibits different chemical shifts. This finding indicates that in the first step of the synthesis, a small part of the dibasic acid is connected to the PA6 segment at both ends; however, it can be clearly seen that the proportion of the link mode is much less and that the effect on the overall molecular chain structure can be ignored. The use of two different dibasic acid blocking agents indicates that this synthetic route can be carried out using aliphatic or aromatic dibasic acids and that the molecular chain design is more flexible and controllable.

The chemical structure of PEA was further verified in FTIR spectroscopy (as shown in Figure 2a). According to the characteristics of the molecular segment design, the characteristic groups (N−H, C=O) are analyzed. The absorption peak at 3304 cm^−1^ is attributed to the stretching vibration absorption peak of the amide (N−H), the bending vibration peaks of the amide (N–H) appear at 1540 cm^−1^ and 680 cm^−1^, and the stretching vibration absorption peak of the amide bond (C=O) is at 1638 cm^−1^. These peaks are characteristic peaks of polyamide 6. The stretching vibration absorption peak of the ester bond (C=O) is at 1737 cm^−1^. As the molecular weight of prePA6 increases, the interval between ester bonds becomes larger, while the characteristic peaks become weaker [26].

Figure 2b shows the spectrum of PEA-p-4k. The C−H bending vibration absorption peak of the 1,4-disubstituted benzene structure can be found at 860 cm^−1^, which indicates that the aromatic dibasic acid can be introduced into the macromolecular chain by this synthetic route.

In the preparation process, the number average molecular weights are controlled between 18,000 and 21,000 by using constant power discharge, and the PDI is between 2 and 2.3. In addition, the oligomer content of PEA obtained by this synthetic route is approximately 2 wt%, which is far below that of PA6 obtained by the hydrolytic polymerization of caprolactam. The test results are shown in Table 2 and Appendix A.

### 3.3. Thermal Properties of PEA

Figure 3 shows the DSC test results of PEA, including the first cooling curves (a) and the second heating curves (b). Other thermal performance values are shown in Table 3. Figure 3a shows that as the molecular weight of prePA6 increases, both the crystallization peak and the melting peak move towards higher temperatures. When the molecular weight of prePA6 reaches 6000, both the melting temperature and the crystallization temperature are close to that of polyamide 6. The thermal properties of polyamide 6 are mainly affected by the molar ratio of the amide group to the methylene group in the main chain [27]. The change in the melting point of PEA is due to the presence of an ester bond in the main chain, which destroys the regularity of the local molecular chain and further reduces the hydrogen bond density. This effect leads to a decrease in the force between macromolecules and reduced crystallization.

There are two kinds of PA6 molecules: cis-aligned and reverse-aligned. The macromolecules only completely combine through hydrogen bonds in the reverse alignment. The three different link modes of the macromolecular chains are arranged in reverse. The influence of hydrogen bonding between the molecular chains is discussed (as shown in Figure 4). The distribution of C=O and N−H in the PEA molecular chain is no longer kept the same as that in the polyamide 6 segment after passing through the ester bond position, and the hydrogen bond structure is destroyed. A molecular chain that does not form a hydrogen bond structure (as shown in the blue frame of Figure 4) may be undergo folding or be combined with other molecular chains to form a hydrogen bond structure. This phenomenon will increase the entanglement of the molecular chain and reduce the regularity of the molecular chain. As the molecular weight of prePA6 gradually increases, the distribution of ester bonds in the segment decreases, reducing the impact on the regularity of the molecular chain and enhancing the degree of crystallization.

The PEA thermal performance test results following polycondensation with different blocking agents and the same molecular weight of prePA6 are shown in Figure 3c,d. When PTA is used as the blocking agent, the melting temperature, crystallization temperature, and crystallinity of PEA are significantly reduced. Due to the introduction of PTA, the phenyl group from PTA enters the molecular chain. The phenyl group has high steric hindrance, increasing the distance and reducing the hydrogen bonding between the molecular chains [28].

The TGA test results of PEA are shown in Figure 5. Figure 5a shows that the higher the molecular weight of prePA6 is, the higher the thermal stability. The initial decomposition temperature (*T_5%_*) of PEA-2k is 367.4 °C, which is still much higher than the processing temperatures of injection molding and melt-spinning. Therefore, these PEA materials have good thermal stability during processing. Figure 5b shows that the thermal decomposition temperatures of PEA with different dicarboxylic acids as the blocking agent were almost identical. The *T_5%_* of PEA-4k and PEA-p-4k are 376 °C and 376.1 °C, respectively. Generally, thermal stability may be improved by the introduction of a phenylated structure into the polymeric backbone. Comparing the thermal stability of PEA-4k and PEA-p-4k, their *T_5%_* are not significantly different. This phenomenon is probably because PTA is added as a capping agent with a low content. In addition, comparing the thermal performances of prePA6 and PEA at the end of the synthesis process shows that they were basically similar. The prePA6-4k and prePA6-p-4k have almost the same molecular chain structure, so PEA-4k and PEA-p-4k have almost the same thermal stability. We can speculate on the thermal properties of PEA by testing the thermal properties of the designed prePA6 molecules.

### 3.4. Crystallization of PEA

Polyamide 6 is a polycrystalline polymer with two different crystal structures: the α type and the γ type. Two diffraction characteristic peaks appear in the α crystal form, and the (200) and (002) crystal form diffraction peaks of α-PA6 are located at 20.1° and 23.8°, respectively. The characteristic γ-PA6 diffraction peak is located at 21.3°. The single-diffraction phenomenon of PA6 is attributed to the characteristic diffraction peaks of (200) and (001) superimposed together [29]. Figure 6a shows that the characteristic peak of PEA is exactly the same as that of PA6, indicating that the crystallinity in PEA is completely determined by the polyamide 6 segment. As the molecular weight of prePA6 increases, the scattering sharpness of the characteristic peak of PEA becomes prominent. The crystal form gradually changes from the γ crystal form to the α crystal form, which is a more stable crystal form. Since the crystallinity of PA6 is mainly dependent on the molar ratio of the amino group to the methylene group, the ester bond in the PEA molecular chain destroys the ordered molar ratio and eventually leads to a decrease in crystallinity. Figure 6b shows that the crystal form of PEA terminated with PTA is mainly the γ crystal form. Compared with the γ crystal form, the α crystal lattice structure is denser. After the introduction of the phenyl group into the molecular chain, the molecular chains become separated from each other, making it more difficult to arrange into the dense α-type lattice of the polyamide 6 chain. The crystalline form of the polyamide 6 segment is more susceptible to transitioning to the metastable γ crystalline form [30].

### 3.5. Tensile Properties of PEA

The mechanical properties of the PEA splines are shown in Figure 7. From Figure 7a,d, as the molecular weight of prePA6 increases, the molecular chains become ordered, and more energy is needed to overcome intermolecular forces to slip the crystal face [31]. Therefore, as the molecular weight of prePA6 increases, the breaking strength and elongation at break gradually improve. The mechanical properties of PEA-6k are close to those of PA6. From Figure 7b,d, the PEA breaking strength (*σ_b_*) is higher than the upper yield point (*σ_su_*). The PEA molecular chains are oriented in the direction of the force, and the molecular chain arrangement is more regular, thereby increasing the crystallinity and improving the mechanical strength.

An interesting phenomenon was found in the text. Figure 6c shows the lower yield point (*σ_sl_*) of the polymer, which is the stress during the yielding process. The lower yield point of PEA-4k is basically the same as that of PA6. As the molecular weight of prePA6 increases, the resistance of the polymer in the yield process becomes higher than that of traditional PA6. The main reason for this is that 8~10 wt % of oligomers are formed during the hydrolysis of caprolactam. For better molding, the PA6 slices need hydrothermal extraction to remove the oligomers, while the PEA samples were processed without hydrothermal extraction. Approximately 2 wt% of the oligomers contained in PEA are mainly composed of cyclic dimers [12]. In PEA, the cyclic dimer is present in the matrix in the form of a rod [32]. In the uniaxial stretching process, the rod crystal is parallel to the stretching direction. It is easy to generate hydrogen bonds with the macromolecular chain, which compensates for a portion of the strength lost due to the decrease in molecular chain regularity (the schematic model is shown in Figure 8). The interaction between the oligomer and the macromolecular chain increases the entanglement between the molecular chains, making mutual movement more difficult. It is manifested in PEA showing an increase in resistance during plastic yielding [33,34]. At the same time, the presence of oligomers also reduces the viscosity during melting and improves the fluidity [35].

### 3.6. Surface Topography of PEA

We unexpectedly discovered the presence of oligomers when SEM was used to observe the cross-section of the material. The PA6 slices before and after hydrothermal extraction and PEA-4k slices without hydrothermal extraction are melted and extruded into round filaments. After liquid nitrogen extraction, the cross-sectional morphology of the samples is observed by SEM, as shown in Figure 9a–c.

Some oligomers can be observed in Figure 9a, which are similar in morphology to the rod-shaped cyclic dimers previously reported in the literature [32]. Most of their directions are the same as the forming direction, and a large amount of oligomers are observed in the cross-section, indicating that too many oligomers will be concentrated in a part of the matrix [32]. This occurrence can lead to stress concentrations that cause the material to appear brittle. There are obvious voids in the cross-section, indicating that the caprolactam will overflow from the matrix during processing, leaving voids to cause defects in the material. This result also indicates the necessity of hydrothermal extraction of the PA6-based polymer before processing. In Figure 9b, it can be seen that PA6 after hydrothermal extraction has a flat profile and no obvious holes or other oligomers. In the PEA-4k sample shown in Figure 9c, only one or several white substances are distributed in the cross-section, similar to the rod-shaped oligomers in Figure 9a. This result proves our speculation on the enhancement in material resistance.

### 3.7. Water Absorption Performance of PEA

The saturated water absorption of PEA slices is shown in Figure 10. The figure shows that PEA is more hydrophilic than PA6. The reason is that water in the free state is mainly present in the amorphous region of the material. The sample with strong crystallization ability does not have enough amorphous areas to retain free water. When phenyl groups are introduced into the molecular chain, the crystallinity of PEA is drastically lowered, causing a large increase in water absorption.

## 4. Conclusions

In this paper, a simple and flexible polymerization route was proposed. The combination of caprolactam hydrolysis polymerization, an esterification reaction, and a transesterification reaction was carried out to prepare a polyamide 6-based copolymer with a molecular structure similar to polyamide 6. PA6 segments of different molecular weights were prepared by changing the type and content of dicarboxylic acid used as a blocking agent. Furthermore, PEA was prepared by esterification and transesterification. PEA prepared by this synthetic route not only retains the excellent properties of PA6 but also increases the possibilities of molecular design during the polymerization process. At the same time, the limitation of the measurement balance of the chemical groups at both ends in the copolymerization modification of polyamide 6 is broken. By ^1^H-NMR and FTIR spectroscopy, it can be observed that adjusting the molecular weight of PA6 is effective in controlling the molecular structure of PEA. In addition, by using PTA as a blocking agent, it was verified that both aliphatic and aromatic dicarboxylic acid can be applied to this polymerization method, which broadens the selectivity of the synthetic raw materials. The effects of prePA6 with different molecular weights and different blocking agents on the molecular structure and material properties of PEA were studied.

The thermodynamic properties, crystal structure, mechanical properties, and water absorption performance of PEA obtained by polycondensation of different prePA6 samples were compared with those of polyamide 6. The results show that as the molecular weight of the PA6 segment continues to increase, the molecular chain regularity increases. When the molecular weight of the PA6 segment reaches 6000, its thermodynamic properties, crystal structure and crystallinity are basically similar to those of PA6. PEA maintains good mechanical properties and water absorption after processing. It is a thermoplastic semi-crystalline polymer with good performance. Moreover, PEA has a low content of oligomers, so it can be processed directly without hydrothermal extraction, which reduces the energy consumption during processing of the PA6-based polymers. It was unexpectedly discovered in this study that a small amount of oligomers also played a certain role in plasticizing, which enhanced the mechanical properties of PEA. This synthetic route can be performed with conventional polyester equipment and has certain potential for industrialization.

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
