# Peer review of "A Novel Synthetic Strategy for Preparing Polyamide 6 (PA6)-Based Polymer with Transesterification"

_polymers, 2019, doi:10.3390/polym11060978_

Reviewer 1 Report

Recommendation: Major revisions.

Comments: The authors synthesized various polyamide 6-based polymers by varying dibasic acids. The authors also compared the proposed method with conventional counterparts, concluding that their proposed method is more flexible and can be valuable for practical applications. This work is interesting. However, there are a few concerns that should be addressed before publication. Hope the authors reflect the following comments and improve the work.

1)       In the keywords section, using “novel synthetic route” as a keyword is not something professional. Instead, please replace with mild term such as “synthetic methodology”.

2)       Is polyamide 6 just Nylon 6? If so, please make a note on it as this is rather a commonly accepted term. In addition, please also include the scientific name for this polymer, which is polycaprolactam.

3)       In Introduction section Page 1 line 38, the term “hydrolysis polymerization” is incorrect. Hydrolytic polymerization is the correct one.

4)       For Page 8 Fig 2, the FTIR is interpreted incorrectly in a way that stretching vibration of C=O in the amide group was labeled as “N-H (amide)”. Please consider the following references for clarification: 1) Molecular-level dispersion of rigid-rod sulfonated aromatic polyamides in epoxy resin for extraordinary improvement in both strength and toughness[J]. Polymer, 2019, 163: 20-28. 2) Highly Biocompatible, Underwater Superhydrophilic and Multifunctional Biopolymer Membrane for Efficient Oil–Water Separation and Aqueous Pollutant Removal[J]. ACS Sustainable Chemistry & Engineering, 2018, 6(3): 3879-3887. 2)

Author Response

Dear Reviewer:

Thank you for your comments on our manuscript, entitled " A Novel Synthetic Strategy for Preparing Polyamide 6 (PA6)-Based Polymer with Transesterification." (ID: 517771) These comments are valuable and useful for modifying and perfecting our papers, and have important guiding significance for our research. We have been carefully studied the comments and made corrections, hoping to get approval. The revised part is marked in red in the paper.

Point 1: In the keywords section, using “novel synthetic route” as a keyword is not something professional. Instead, please replace with mild term such as “synthetic methodology”.

Response 1: As the reviewer said using “novel synthetic route” as a keyword is not something professional. We think that the keywords given by the reviewers are more representative, so this part was rewritten based on the Reviewer's recommendations.

Point 2: Is polyamide 6 just Nylon 6? If so, please make a note on it as this is rather a commonly accepted term. In addition, please also include the scientific name for this polymer, which is polycaprolactam.

Response 2: Yes, it is. The Reviewer's comments are very pertinent, and we should include some of the commonly names for polyamide 6 to better understand our paper. We have already added this part to the Introduction.

Point 3: In Introduction section Page 1 line 38, the term “hydrolysis polymerization” is incorrect. Hydrolytic polymerization is the correct one.

Response 3: We are very sorry for our mistakes in writing. We have made correction according to the Reviewer’s comments.

Point 4: For Page 8 Fig 2, the FTIR is interpreted incorrectly in a way that stretching vibration of C=O in the amide group was labeled as “N-H (amide)”. Please consider the following references for clarification: 1) Molecular-level dispersion of rigid-rod sulfonated aromatic polyamides in epoxy resin for extraordinary improvement in both strength and toughness[J]. Polymer, 2019, 163: 20-28. 2) Highly Biocompatible, Underwater Superhydrophilic and Multifunctional Biopolymer Membrane for Efficient Oil–Water Separation and Aqueous Pollutant Removal[J]. ACS Sustainable Chemistry & Engineering, 2018, 6(3): 3879-3887. 2)

Response 4: Response 4: Considering the Reviewer’s suggestion, we read the references you provided to us and compared them to the corresponding content in our manuscript. We think that the label “N-H(amide)” in Page 8 Fig 2 is a reasonable structural explanation. In Fig.2 the bending vibration peaks of the amide (N-H) appear at 1540 cm-1 and 680 cm-1, and the stretching vibration absorption peak of the amide bond (C=O) is at 1638 cm-1. These peaks are characteristic peaks of polyamide 6. Please consider the following references for clarification: 1) Yuan, R.; Fan, S.; Wu, D.; Wang, X.; Yu, J.; Chen, L.; Li, F., Facile synthesis of polyamide 6 (PA6)-based thermoplastic elastomers with a well-defined microphase separation structure by melt polymerization. Polymer Chemistry 2018, 9 (11). 2) Huang, J.; Lan, J.; Lin, S.; Li, G.; Gu, L., Synthesis and nonisothermal crystallization kinetics of thermoplastic polyamide6 elastomers. Journal of Applied Polymer Science 2019, 136 (14), 47388. 3) Kong W , Yang Y , Liu Z , et al. Structure-property relations of novel polyamide-6 elastomers prepared through reactive processing[J]. Journal of Polymer Research, 2017, 24(10):168.

If you have any better comments for our manuscript, please feel free to write to me again. We are very grateful for your comments on our paper.

Thank you and extend your sincere regards.

Kind regards,

Shengming Zhang

Reviewer 2 Report

Aim of this manuscript is the development of a novel strategy for polyamide6(PA6)_based polymer synthesis. The manuscript is interesting and results are well conceived and discussed. I have only some suggestions:

1)      Raw 77 add PET the first time you mention polyethylene terephthalate.

2)      The frequency of the NMR spectrometer is not a vibration frequency, please correct in the NMR paragraph.

3)      On scheme 1 at step 3 the reaction conditions have to be added for clarity.

4)      On table 1 MHNMR is incorrect; the exact symbol is Mn

5)      Formula 3: you have to explain all terms (also 113 and 146)

6)      When compared, the integral ratio of the ethylene glycol segment in the proton spectra a and b the value is not significantly reduced, changes from 094 to 0.85.  Can you explain what means?

7)      In figure 1c, the aminocaproic hydrogens are down-field shifted.

8)      Generally, thermal stability may be improved by the introduction of a phenylated linkage into the polymeric backbone. Please add some comments.

Author Response

Dear Reviewer:

Thank you for your comments on our manuscript, entitled " A Novel Synthetic Strategy for Preparing Polyamide 6 (PA6)-Based Polymer with Transesterification." (ID: 517771) These comments are valuable and useful for modifying and perfecting our papers, and have important guiding significance for our research. We have been carefully studied the comments and made corrections, hoping to get approval. The revised part is marked in red in the paper.

Point 1: Raw 77 add PET the first time you mention polyethylene terephthalate..

Response 1: We are very sorry for our negligence of mark its abbreviation when polyethylene terephthalate firstly proposed. We have already supplemented the polyethylene terephthalate abbreviation in the corresponding position in the manuscript.

Point 2: The frequency of the NMR spectrometer is not a vibration frequency, please correct in the NMR paragraph.

Response 2: We are sorry for the lack of a deep understanding of the NMR test principle. We have made correction according to the Reviewer’s comments.

Point 3: On scheme 1 at step 3 the reaction conditions have to be added for clarity.

Response 3: We lacked some of the necessary reaction conditions on scheme 1 at step 3. This will cause some confusion when reading this part. We replenish reaction conditions such as temperature and vacuum in scheme 1 at step 3.

Point 4: On table 1 MHNMR is incorrect; the exact symbol is Mn.

Response 4: We are very sorry for our mistakes in writing. We have made correction according to the Reviewer’s comments.

Point 5: Formula 3: you have to explain all terms (also 113 and 146)

Response 5: We lacked some of the necessary explanation of the number (113 and 146) on formula 3. This will cause some confusion when reading this part. We add the explain of the different numbers after the formula 3.

Point 6: When compared, the integral ratio of the ethylene glycol segment in the proton spectra a and b the value is not significantly reduced, changes from 094 to 0.85. Can you explain what means?

Response 6: In the synthetic of prePA6, in order to increase the esterification rate, the ethylene glycol is used in excess. Therefore, in the atmospheric esterification process, a nitrogen gas stream is used to carry out water and ethylene glycol which is not involved in the reaction. Finally, the reaction is balanced. We believe that the effect of ethylene glycol monomer on NMR spectra in prePA6 is negligible. After the Step 3, the integral ratio of the ethylene glycol segment in the proton spectra a and b changes from 0.94 to 0.85, it corresponds to a 10% reduction in ethylene glycol. This value is lower than the theoretical calculation. This is because some small molecules such as diethylene glycol are inevitably formed in the polymerization of polyester. The same reactions can also occur in the synthetic strategy of our manuscript. More importantly, these small molecules are also linked to the macromolecule by transesterification. The ethylene glycol segments are combined in the NMR spectrum, so this part will also be calculated. Therefore, we cannot directly compare the reduction of ethylene glycol with the theoretical value. In our manuscript, we only hope to certify that the molecular weight increase is achieved by transesterification. And we consider that ethylene glycol reduction of 10% has enough persuasion.

Point 7: In figure 1c, the aminocaproic hydrogens are down-field shifted.

Response 7: Considering the Reviewer’s comment, we re-examined the NMR spectrum and corresponding interpretation in Fig. 1(c). We realized that we have an obvious identification error with δ1' on the NMR spectrum. We mistakenly mark the δ1' to the hydrogen on the carbon that is linked to the amino group in the macromolecular chain. We added the revised Fig. 1(c) and explanation to the manuscript. As the same as Fig. 1(a), the δ1' is the hydrogen on the carbon that is linked to the ester bond.

Point 8: Generally, thermal stability may be improved by the introduction of a phenylated linkage into the polymeric backbone. Please add some comments.

Response 8: Considering the Reviewer’s comment, we have added some comments on the thermal stability of PEA after the introduction of phenylated structure. Comparing the thermal stability of PEA-4k and PEA-p-4k, their T5% are not significantly different. This phenomenon is probably because PTA is added as a capping agent (this is certify by NMR) with a low content (prePA6-p-4k feed ratio of CPL to PTA is 34:1). In addition, Comparing the thermal performances of prePA6 and PEA at the end of the synthesis process shows that they were basically similar. The prePA6-4k and prePA6-p-4k have almost the same molecular chain structure, so PEA-4k and PEA-p-4k have almost the same thermal stability. Adding more PTA or changing to other benzene series will change PEA thermal stability. This will also be a research focus of our future work.

If you have any better comments for our manuscript, please feel free to write to me again. We are very grateful for your comments on our paper.

Thank you and extend your sincere regards.

Kind regards,

Shengming Zhang

Round  2

Reviewer 1 Report

Accept